# Magmatic Processes of the Upper Cretaceous Susuma–Nagaho Plutonic Complex, Southwest Japan: Its Role on Crustal Growth and Recycling in Active Continental Margins

**Shogo Kodama** [1,2], **Masaaki Owada** [1,*], **Mariko Nagashima** [1] and **Atsushi Kamei** [3]

1 Division of Earth Sciences, Graduate School of Science and Technology for Innovation, Yamaguchi University, Yamaguchi 753-8512, Japan; shogo_kodama@taiheiyo-c.co.jp (S.K.); nagashim@yamaguchi-u.ac.jp (M.N.)
2 Taiheiyo Consultant Co., Ltd., West Japan Technology Department, Sanyoonoda 756-0817, Japan
3 Department of Earth Sciences, Graduate School of Natural Science and Technology, Shimane University, Matsue 690-8504, Japan; kamei-a@riko.shimane-u.ac.jp
* Correspondence: owada@yamaguchi-u.ac.jp

**Abstract:** Magmatic processes in the active continental margins are one of the important issues to understand the evolution of the continental crust. The Cretaceous Susuma–Nagaho plutonic complex, southwest Japan, is situated at the continental arc, and made up of gabbro, quartz diorite to granodiorite, and granite. According to the field occurrence, they are coeval intrusive rocks, and the biotite K–Ar ages of the granodiorite and granite are approximately 93 Ma, corresponding to the period of a magmatic flare-up in southwest Japan. Based on the whole-rock chemical analyses including Sr–Nd isotopic compositions, the granodiorite magma has been formed through fractional crystallization of basaltic magmas, whereas the origin of granite magma involved partial melting of the continental crust. The gabbro contains calcium-rich plagioclase (An > 90) and the presence of early crystallized hornblende, indicating its derivation from a hydrous basaltic magma. Such basaltic magma intruded into the middle to lower crust and supplied the heat energy necessary for crustal partial melting and granitic magma formation. The fractional crystallization and crustal melting took place at the same time, playing an important role in the crustal growth and differentiation during the magmatic flare-up event.

**Keywords:** fractional crystallization; basaltic magma; crustal anatexis; crustal evolution; Susuma–Nagaho plutonic complex (SNPC)





## 1. Introduction

Mantle-derived basaltic magmas are important materials for the growth of continental crust, although the upper crust of continental magmatic arcs is composed mainly of granite. However, the granitic magma cannot be directly produced by the partial melting of the mantle because it is not in equilibrium with mantle peridotite [1]. The crust-forming granitic rocks, therefore, are thought to be generated by the following two mechanisms: (1) differentiation through processes of fractional crystallization with or without crustal assimilation of mantle-derived basaltic magma and (2) partial melting of meta-igneous and/or meta-sedimentary rocks. The former process contributes to the growth of continental crust, whereas the latter mechanism proceeds the chemical differentiation of crust without any addition to crustal materials from the mantle. Therefore, unraveling the respective contribution of two different mechanisms in the formation of granitic suites is necessary to understand the evolution of continental crust [2,3].

The volcano–plutonic complexes associated with the subduction of the Izanagi plate during the Cretaceous to Paleogene were generated as the magmatic flare-up in the Inner Zone of southwest Japan. The igneous complex in this zone consists mainly of granitoids. In terms of their lithologies, intrusive ages, and associated ore deposits, the granitoids can

be divided into three belts: the San-in, San-yo, and Ryoke belts from north to south [4] (Figure 1a). The voluminous granitoids are accompanied by small amounts of gabbro and diorite stocks as coeval intrusive rocks due to field occurrence between granitic and mafic rocks. In other words, magmatism from the Cretaceous to the Paleogene could have promoted both growth and differentiation of the Earth's crust.

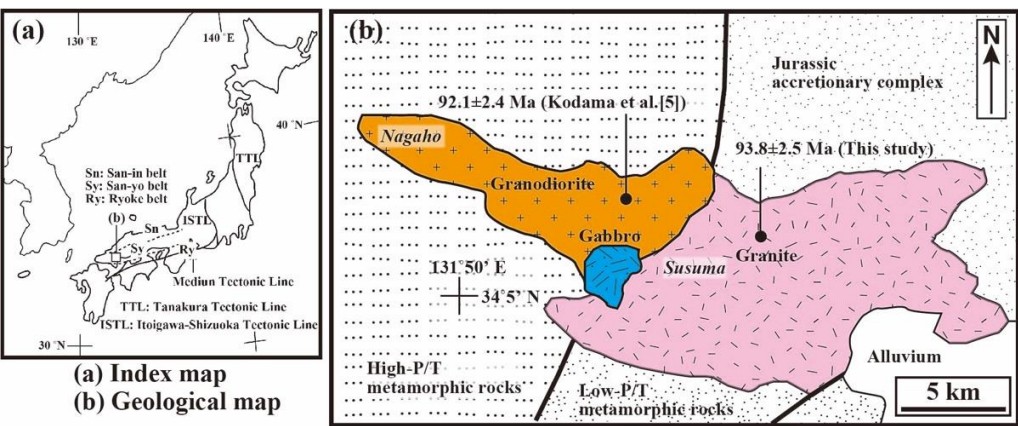

**Figure 1.** Geological map of the study area. (**a**) Index map with the San-in, San-yo, and Ryoke Belts. (**b**) Geological map of study area (after reference [5]). Sample sites for biotite K–Ar dating are also marked.

The Cretaceous gabbro–granodiorite–granite suite is exposed in the Susuma–Nagaho area, the western part of the San-yo belt, west Chugoku district, southwest Japan [5]. In this paper, these plutonic rocks are referred to as the Susuma–Nagaho Plutonic Complex, SNPC hereafter. We address petrography, and mineral and whole-rock geochemical analyses to reveal their petrological characteristics and the petrogenesis of the SNPC by the highlighting following three points: (1) relationships between gabbro and granodiorite, (2) petrogenesis of granitic magma, and (3) the origin of basaltic magma and mantle dynamics. The SNPC provides an opportunity for the magmatic processes and their role in crustal architecture. This study essentially contributes to the formation of continental magmatic arcs and mantle dynamics above the subducting oceanic plate. The mineral and whole-rock chemical compositions of the SNPC except for the granite are quoted from the published paper [5].

## 2. General Geology

The western end of Honshu Island is underlain by the high-pressure type of metamorphic rocks and the accretionary complexes as pre-Cretaceous rocks, and the Cretaceous igneous complex with low-pressure type metamorphic rocks, known as the Ryoke plutono–metamorphic complex. The SNPC shows an elongated body that is more than 25 km in the E–W direction and 10 to 15 km in the N–S direction (Figure 1b). The SNPC intrudes the high-pressure type Suo metamorphic rocks of the Triassic age, the Kuga group that is a Jurassic accretionary complex, and the low-pressure type Ryoke metamorphic rocks with Cretaceous age (Figure 1b).

The SNPC consists mainly of various kinds of lithologies: gabbro, quartz diorite, granodiorite, and granite. Based on the occurrence and petrography, the SNPC is classified into three groups, gabbro, quartz diorite to granodiorite, and granite groups. Although the quartz diorite to granodiorite group is made up of diorite, quartz diorite, and granodiorite, the lithology of granodiorite is dominant. In this paper, we refer to this group as "granodiorite". The granite and granodiorite groups occur as relatively large plutons, whereas the gabbro group appears as a small stock (1 × 1 km, Figure 1b) and enclaves or syn-plutonic dikes in the granite and granodiorite groups. The gabbro can be subdivided into fine-grained porphyritic gabbro (Fn-Gab) and coarse-grained cumulus gabbro (Co-Gab) (Figure 2a,b). Although Fn-Gab and Co-Gab generally show massive and dark colors,

Co-Gab locally shows a layered structure (Figure 2b). Figure 2c shows the occurrence of the granodiorite (left half) and granite (right half). The granodiorite is pale gray in color and medium-grained without any foliations. The granite shows massive and coarse-grained and leucocratic rather than the granodiorite. The boundary between the granodiorite and granite is not in sharp contact with intricate shapes (Figure 2c). The granite locally includes xenoliths of pelitic schists with weak foliations derived from the Ryoke metamorphic rocks (Figure 2d). The granodiorite and granite include the mafic magmatic enclaves (MME) probably derived from the gabbro stock. The gabbro locally bears the granodiorite inclusions indicating mutual intrusion relationships between the gabbro and the granodiorite. The straddled crystal such as an alkali-feldspar appears in the boundary between the gabbro and the granite.

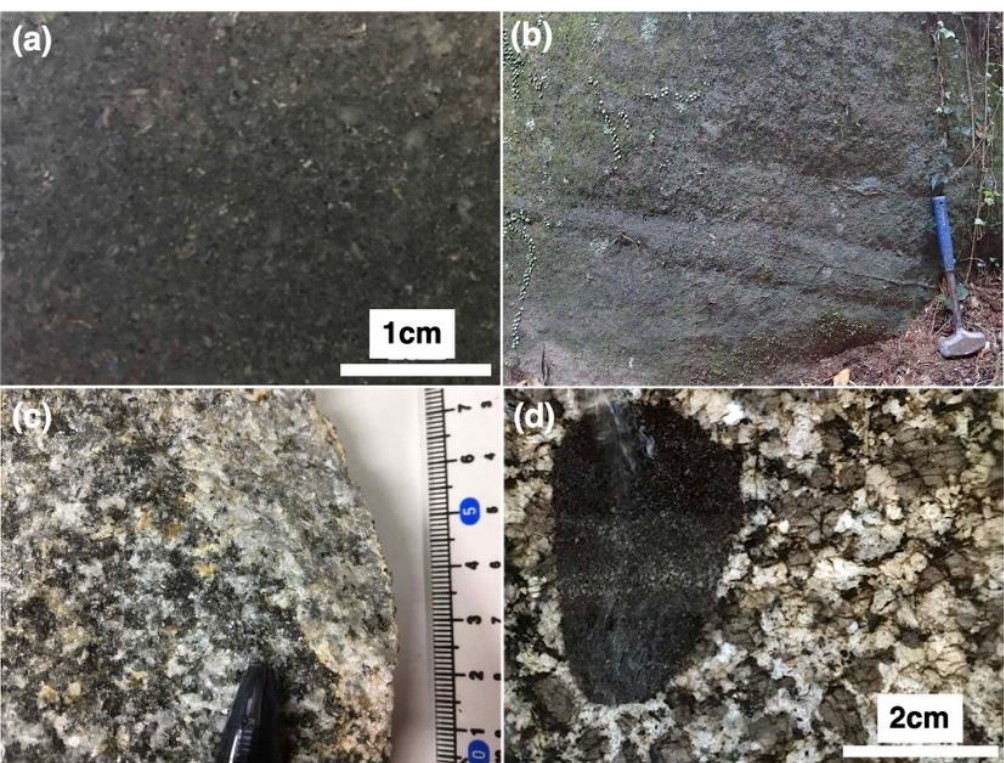

**Figure 2.** Photographs including field occurrence of the studied samples. (**a**) Photograph of Fn-Gab. (**b**) Co-Gab showing layered structure. (**c**) Mode of occurrence of the granodiorite (left hand) and granite (right hand). The boundary between the granodiorite and granite is not in sharp contact with intricate shape. (**d**) The granite including a pelitic schist as a xenolith.

We separated the biotite from the granodiorite and granite for K–Ar dating. The rock sample was crushed by tungsten mortar. After a cloth sieve was used to reduce the particle size to 0.12–0.20 mm, the biotite was concentrated in the electromagnetic separator at Yamaguchi University. The K–Ar dating was conducted by GeochronEx Ltd. The age was calculated by using decay constants $\lambda_\beta$ = 0.4962/Ga and $\lambda_\varepsilon$ = 0.0581, and the isotopic abundance ratio $^{40}K/K$ = 0.0001167. Such an analytical procedure was described in the published paper [5]. The biotite K–Ar dating of the granodiorite and granite gave the ages of 92.1 ± 2.4 Ma for the granodiorite [5] and 93.8 ± 2.5 Ma for the granite (this study, Table 1, Figure 1b), respectively. According to their occurrence and the results of K–Ar dating, the granodiorite, granite, and gabbro simultaneously intrude on each other and would experience the same cooling processes.

**Table 1.** The results of biotite K–Ar age of the granite.

| Sample 17101203 | Mineral | K (%) | $^{40}$Ar rad (nl/g) | $^{40}$Ar air (%) | Age (Ma) | Error (1s) |
|---|---|---|---|---|---|---|
| Granite | Biotite | 7.11 | 26.11 | 6.2 | 93.8 | 2.5 |

## 3. Description and Mineral Chemistry of Analyzed Samples

The modal compositions of the granodiorite and granite were obtained by point counting under the microscope using thin sections. At least 2000 points were counted for each sample. The results are shown in the quartz (Qz)–alkali-feldspar (Afs)–plagioclase (Pl) diagram (Figure 3). According to Figure 3, the granodiorite and granite are plotted within the field of quartz diorite to granodiorite and tonalite to granite, respectively. We also conducted the modal analyses of Co-Gab. Co-Gab is divided into two lithofacies in terms of modal values of plagioclase and mafic minerals. The modal values of plagioclase and mafic minerals show 70–73 and 48–49% and 26–30 and 50–52% for the leucocratic and melanocratic facies, respectively. The Co-Gab of melanocratic and leucocratic facies refers as CM-Gab (coarse-grained melanocratic gabbro) and CL-Gab (coarse-grained leucocratic gabbro) hereafter.

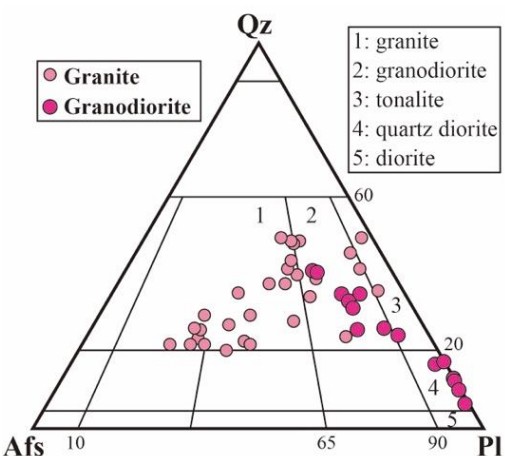

**Figure 3.** Modal values of the granodiorite and granite plotted in quartz (Qz)–alkali feldspar (Afs)–plagioclase (Pl) diagram. The granodiorite shows lithological variations from quartz diorite to granodiorite. The granite is mostly plotted within the granite field, but some samples show tonalite to granodiorite compositions.

Photomicrographs of studied samples are shown in Figure 4. Mineral compositions were determined using a JXA-8230 electron probe micro-analyzer (EPMA) at the Center for Instrumental Analyses, Yamaguchi University. Operating conditions for chemical analyses were an accelerating voltage of 15 kV, a specimen current of 20 nA, and a beam diameter of 1–5 micro-meters. Wavelength-dispersion spectra were collected with LiF, PET, and TAP crystals to identify interfering elements and located the best wavelengths for background measurements. The ZAF method was used for data correction. Under the conditions described, analytical errors are ± 2% for major elements as estimated from the reproducibility observed in multiple measurements. Mineral chemistries of the granite are listed in Supplementary Table S1. Analytical data are plotted in Figure 5.

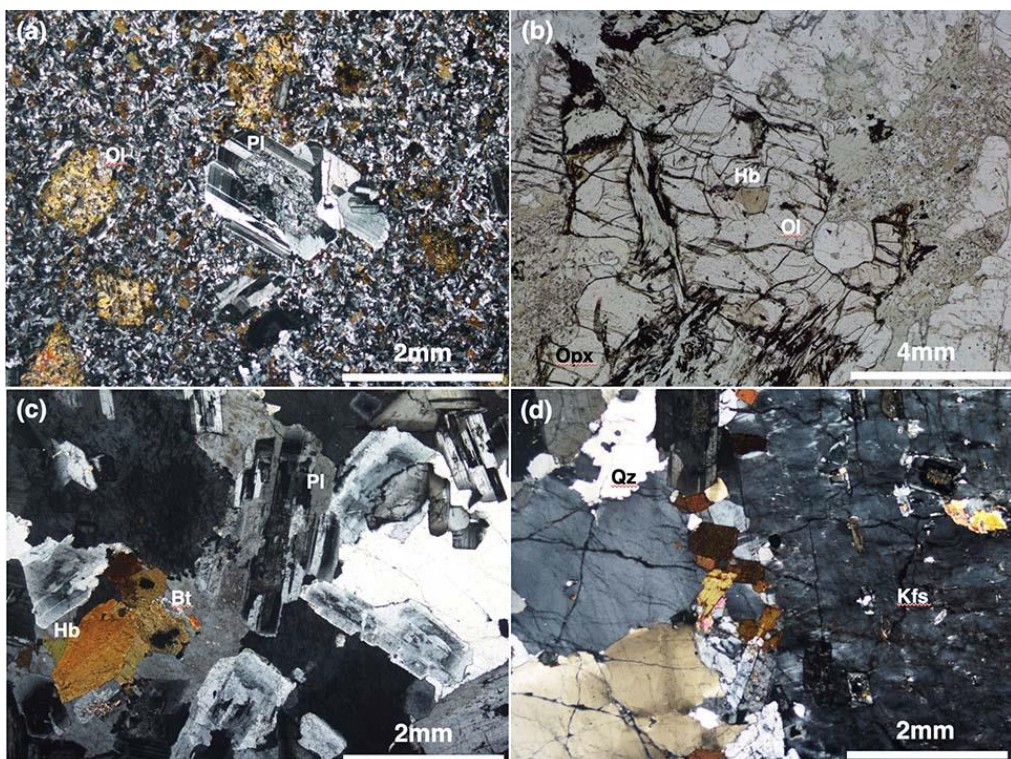

**Figure 4.** Photomicrographs of the study samples. (**a**) Fn-gabbro. (**b**) CM-gabbro. (**c**) Granodiorite. (**d**) Granite. Pl: plagioclase, Qz: quartz, Kfs: K-feldspar, Ol: olivine pseudomorph replaced by fine-grained minerals probably smectite, Opx: orthopyroxene, Hb: hornblende, Bt: biotite.

Fn-Gab shows porphyritic texture and consists mainly of plagioclase, clinopyroxene, hornblende, and olivine pseudomorph as phenocrysts (Figure 4a). The occurrence of Fn-Gab in Figure 4a is unknown, but it is likely to be a syn-plutonic dike because it is exposed on the geological map in the peripheral area of the gabbro group and shows the fine-grained porphyritic texture. A groundmass includes plagioclase, hornblende, biotite, and ilmenite. The anorthite contents [An = 100 × Ca/(Ca + Na + K)] of plagioclase phenocrysts show a wide compositional range, An = 70 to 80 for core and An = 42 to 68 for rim (Figure 5a). The olivine phenocrysts are completely replaced by aggregates of fine-grained minerals that are probably smectite (Figure 4a). Al ions at the tetrahedral site ($Al^{IV}$) are up to 1.5 apfu for the hornblende phenocrysts (Figure 5a). Na + K (atomic per formula unite, apfu) has a wide range from 0.1 to 0.7. Mg# [100 × Mg/(Mg + Fe)] vs. Al (apfu) of biotite are shown in Figure 5c. Mg# ranges from 21 to 48 but Al is almost constant around 1.0 apfu.

CM-Gab consists of plagioclase, hornblende, clinopyroxene, orthopyroxene, and olivine with small amounts of biotite (Figure 4b). Ilmenite, magnetite, and chromian spinel are present as accessory minerals. The axial color of the hornblende is variable. Brown hornblende grains occur as euhedral to subhedral shapes and are locally included in olivine (Figure 4b) and orthopyroxene. Green to pale green tremolitic hornblende grains are present on the rim parts of the brown hornblende and/or those of pyroxene grains (Figure 4b). The An contents of plagioclase generally show more than 90 (Figure 5a). The brown hornblende has the Al-enriched core, in which Al ions at the tetrahedral site ($Al^{IV}$) attain more than 2 atoms apfu, whereas the green hornblende exhibits $Al^{IV}$ less than 1.5 apfu (Figure 5b). Biotite in CM-Gab is the highest Mg# among the SNPC from 76 to 78 with 2.6 to 3.1 in Al (apfu) (Figure 5c).

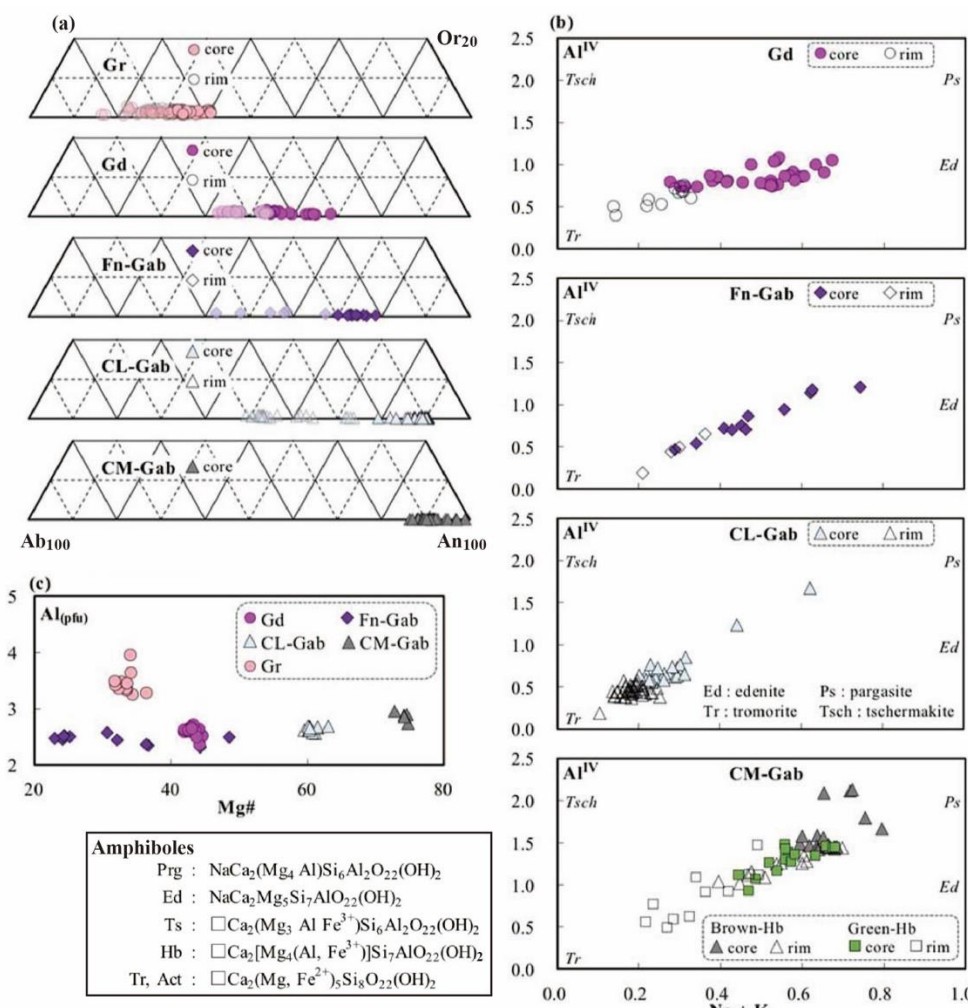

**Figure 5.** Chemical composition of each mineral. (**a**) plagioclase, (**b**) hornblende, (**c**) biotite. Gd, granodiorite; Gr, granite. See text for a detailed explanation.

CL-Gab is composed of plagioclase and hornblende with small amounts of biotite and quartz. Olivine and pyroxenes are locally included within hornblende. Hornblende shows anhedral and interstitial occurrence, but its grain size is larger than plagioclase. Plagioclase possesses a wide core with An = 79 to 91 and a thin rim with An = 50 to 74 (Figure 5a) and occurs as a poikilitic inclusion in hornblende. Forsterite contents (Fo = 100 × Mg/Mg + Fe) of olivine in both CM-Gab and CL-Gab are less than 75 (not shown in Figure 5). Maximum $Al^{IV}$ is 1.8 slightly less than that of the brown hornblende in CM-Gab. Mg# and Al (apfu) of biotite show 59–61 and 3.0–2.5, respectively (Figure 5c). Olivine and quartz are not in direct contact, and quartz occurs interstitially with euhedral to subhedral minerals.

The granodiorite consists mainly of plagioclase, quartz, biotite, and hornblende with a small amount of alkali-feldspar (Figure 4c). Titanite, ilmenite, apatite, and zircon are accessory minerals. Plagioclase has An = 52 to 68 for core parts and An = 43 to 54 for rim parts (Figure 5a). The hornblende shows variable axial color with brown and pale brown to greenish brown. Modal compositions of hornblende and biotite vary in each sample. The chemical compositions of hornblende show a gradually changing edenitic core to the actinolitic rim with $Al^{IV}$ less than 1.2 apfu (Figure 5b). Mg# of biotite ranges from 41 to 45 with 2.4 to 2.8 Al apfu (Figure 5c)

The granite is characterized by alkali-feldspar phenocrysts and the matrix includes plagioclase, quartz, and biotite (Figure 4d). Muscovite and garnet are locally present. Ilmenite, zircon, and apatite are included as accessory minerals. The An contents of

plagioclase possess 27 to 40 in the core and 16 to 30 in the rim (Figure 5a). Mg# of biotite ranges from 32 to 38. The aluminum ion of biotite shows the highest values among the SNPC ranging from 3.2 to 4.1 apfu (Figure 5c).

## 4. Whole-Rock Geochemistry

We performed the whole-rock chemical analyses for the granite because the chemical compositions of the granodiorite and gabbro have been reported by the published paper [5]. Geochemical data will be described in the next section along with previously published data [5].

### 4.1. Analytical Procedure

Geochemical analysis, for determining major and trace elements, was performed via X-ray fluorescence (XRF) spectrometry at the Center for Instrumental Analyses, Yamaguchi University. All analyses were made on glass beads using an alkali flux comprising lithium tetraborate. The analytical procedure is described in the published paper [6]. Trace elements including rare earth elements (REEs) were determined with inductively coupled plasma mass spectrometry (ICP-MS) at Activation Laboratory Ltd., Canada. The results of major and trace elements of the granite are shown in Supplementary Table S2.

The Sr and Nd isotopic analyses of the granite were performed using the following procedures. Sr and Nd from the powdered whole-rock samples were extracted at the Yoshida Research and Education Building of Yamaguchi University by the following methods [7]. Isotopic analyses were conducted via thermo-ionization mass spectrometry (TIMS, Finigan MAT-262) at the Department of Earth and Environment Sciences, Shimane University. The detailed analytical procedure is described in the published paper [8]. Measured $^{87}Sr/^{86}Sr$ and $^{143}Nd/^{144}Nd$ ratios were normalized to $^{86}Sr/^{88}Sr = 8.357209$ and $^{146}Nd/^{144}Nd = 0.7219$, respectively. The $^{87}Sr/^{86}Sr$ and $^{143}Nd/^{144}Nd$ ratios were corrected relative to reference values of 0.710242 (NIST SRM 987 [9]) and 0.511858 (JNdi-1 [10]), respectively. The Rb, Sr, Sm, and Nd concentrations were measured by ICP-MS. The initial Sr and Nd isotope ratios were calculated using the decay constants $\lambda^{87}Rb = 1.42 \times 10^{-11}$/year [11] and $\lambda^{147}Sm = 6.54 \times 10^{-12}$/year [12]. The epsilon values ($\varepsilon$) were calculated using the following method [13]. Considering the results of biotite K–Ar dating of the granodiorite (92 Ma [5]) and the granite (94 Ma; this study), the Sr and Nd isotopic ratios were corrected to 93 Ma, because no zircon U–Pb ages were reported for the rocks in the SNPC. The results of Sr and Nd isotopic analyses of the granite are listed in Supplementary Table S3.

### 4.2. Results

The variation diagrams with $SiO_2$ vs. major and trace elements are shown in Figure 6. The granodiorite possesses a wide compositional range with $SiO_2$ contents from 52 to 72 wt% because this group includes lithological variation from quartz diorite to granodiorite. This is confirmed by chemical data (see also Figure 3). The $SiO_2$ contents of Fn-Gab and Co-Gab, however, show 43 to 52 wt% and 38 to 50 wt%, respectively. The granite has 67 to 78 wt% in $SiO_2$. The analyzed samples mostly belong to the subalkaline series in the total alkaline ($Na_2O + K_2O$ wt%) vs. $SiO_2$ wt% diagram. The gabbro and granodiorite are almost metaluminous but the granite shows peraluminous features in terms of the alumino-saturation index [$Al_2O_3/(CaO + Na_2O + K_2O)$: mole ratio]. The granodiorite and Fn-Gab make continuous trends on the variation diagrams; however, Co-Gab including both CL-Gab and CM-Gab is scattered around Fn-Gab. Some samples of Fn-Gab show $MgO = 10$ wt%. The MgO contents decrease with increasing $SiO_2$. The granodiorite and granite make different trends in $SiO_2$ wt% vs. Sr and Zn ppm diagram.

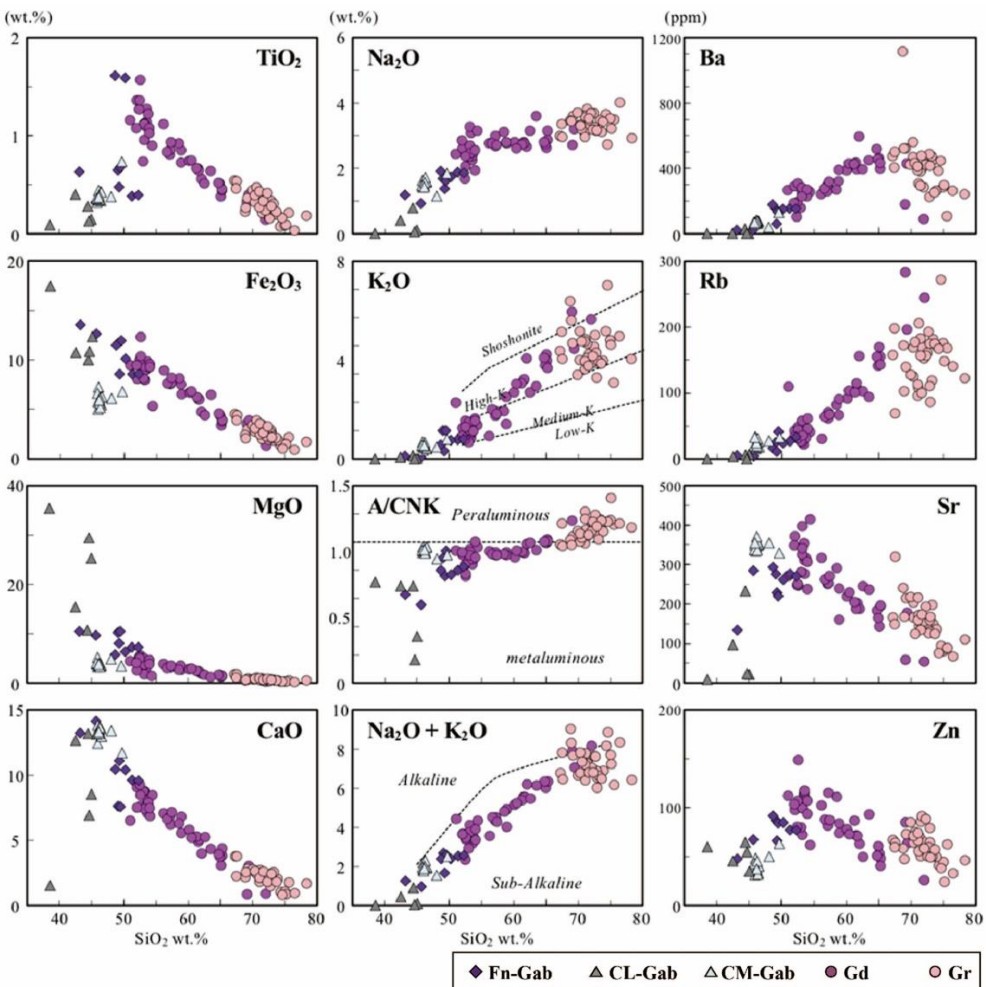

**Figure 6.** Variation diagrams of the SNPC. Gd, granodiorite; Gr, granite. The granodiorite and granite show different trends in $SiO_2$–Sr and –Zn diagrams.

The spider diagrams normalized to the value of primitive mantle [14] and the chondrite normalized REE patterns are shown in Figure 7. The analyzed samples show enrichment of LIL elements and depletion of HFS ones with remarkably negative Nb and Ta anomalies in the spider diagrams (Figure 7b). The chondrite-normalized REE patterns for the analyzed samples are of enriched light-REE (LREE) without two samples of CM-Gab but show almost flat patterns in heavy-REE (HREE). The granodiorite and granite show similar REE patterns with negative Eu anomalies (Figure 7a). In contrast, the gabbro exhibits slightly positive Eu anomalies (Figure 7a).

The initial epsilon-Sr ($\varepsilon$SrI) and -Nd ($\varepsilon$NdI) values calculated at 93 Ma are shown in the epsilon diagram (Figure 8). The granodiorite, Fn-Gab, CM-Gab, and CL-Gab have similar isotopic compositions. On the other hand, the granite shows low-$\varepsilon$NdI values rather than the granodiorite and gabbro although the Sr–Nd isotopic compositions of the granite are plotted in the same field as the Ryoke granitoids. It is noteworthy that the REE patterns of granodiorite and granite are similar to each other, but the Sr–Nd isotopic compositions are different. Moreover, all samples described here possess the negative $\varepsilon$NdI values.

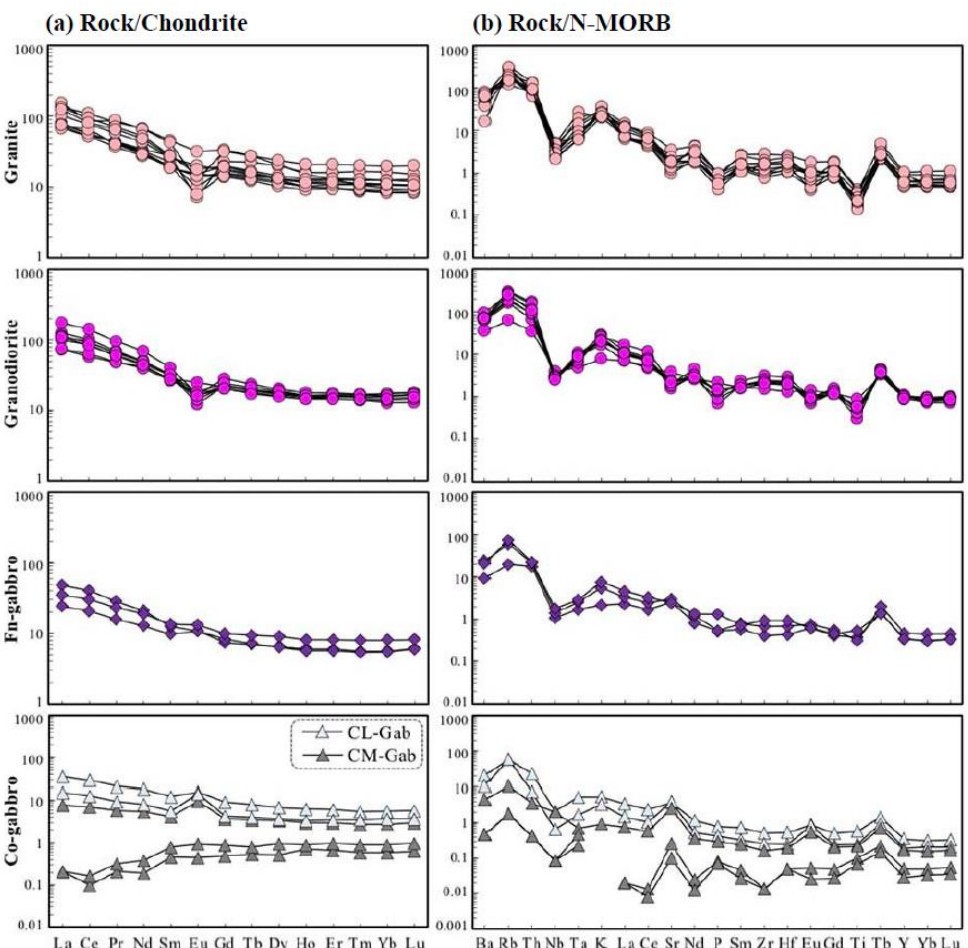

**Figure 7.** Chondrite normalized REE patterns (**a**) and spider diagram normalized to N-MORB (**b**) of the SNPC. Normalized values of C1 chondrite and N-MORB are after the published paper [14].

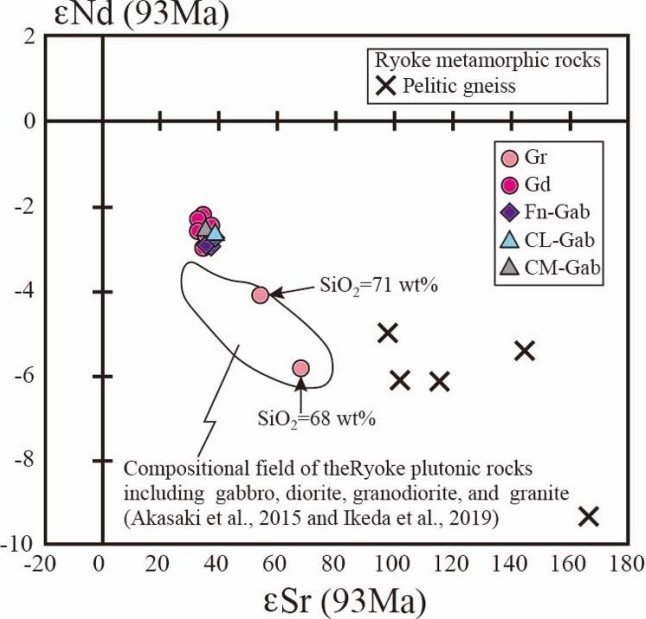

**Figure 8.** Epsilon Sr and Nd isotopic initial values corrected with 93Ma of the SNPC. Gr, granite [15]; Gd, granodiorite [16].

## 5. Discussion

### 5.1. Genetic Relationships between Gabbro and Granodiorite

The gabbro contains hornblende with various compositions (Figure 5b). The brown hornblende occurring as poikilitic inclusions within orthopyroxene and olivine in CM-Gab has high $Al^{IV}$ values of more than 2.0 apfu (Figure 5) and shows relatively high-Mg values (Mg# = 65 to 76). The hornblende having the highest $Al^{IV}$ values would, therefore, be crystallized close to a liquidus, whereas the low-$Al^{IV}$ green hornblende was formed in the differentiated magma under the state of cooling. Pressure and temperature conditions during crystallization of both high- and low-$Al^{IV}$ hornblende grains in CM-Gab were estimated using the hornblende geothermobarometer [17,18]. The results are shown in Figure 9. This figure also plots the emplacement conditions of the granodiorite using the hornblende–plagioclase geothermometer [18] and geobarometer [19].

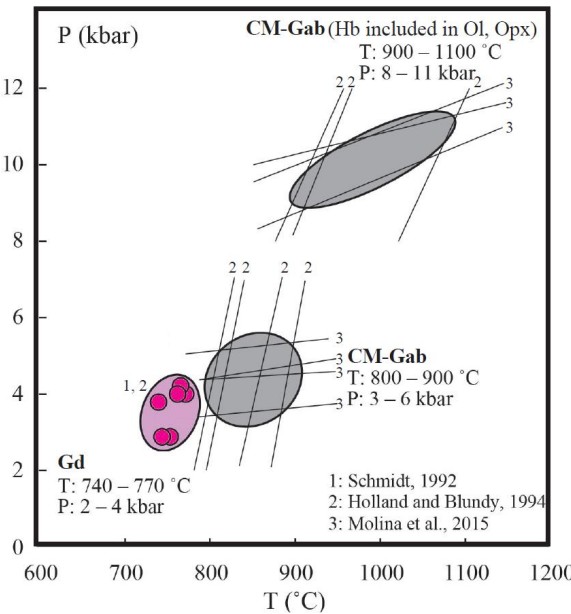

**Figure 9.** P–T diagram showing crystallization conditions for CM-Gab and the granodiorite. Pressure and temperature conditions of CM-Gab and the granodiorite adopted methods of hornblende geobarometer and plagioclase-hornblende geothermometer [17–19]. Gd, granodiorite.

The hornblende occurring as inclusions within olivine and orthopyroxene with high $Al^{IV}$ and Mg values was crystallized at the lower crust conditions between 8 and 11 kbar, whereas the other hornblende was formed at the middle to upper crust conditions (3–6 kbar) (Figure 9). In other words, the basaltic magma crystallized hornblende with various depths from lower to upper crust conditions during the time for magma ascending, and the magma finally emplaced at the shallow level at 3–6 kbar. The hornblende in the granodiorite was crystallized at pressure conditions from 2 to 4 kbar. The emplacement depth of the granodiorite, therefore, is similar to that of CM-Gab (Figure 9).

The granodiorite and Fn-Gab geochemically make trends in the variation diagrams (Figure 6). Co-Gab locally shows layered structures (Figure 2b). CL-Gab and one sample of CM-Gab have positive Eu anomalies in the chondrite-normalized REE patterns, and the granodiorite shows negative Eu anomalies (Figure 7a). Fn-Gab, however, shows no Eu anomalies (Figure 7a). In addition, the initial Sr and Nd isotopic compositions of the granodiorite, Fn-Gab, and Co-Gab are plotted within the same field (Figure 8). Considering these lines of petrological data, the granodiorite and the gabbro would have originated from a single source [5]. The granodiorite magma would have differentiated from Fn-Gab magma through the process of fractional crystallization leaving plagioclase and olivine, pyroxenes, and hornblende consisting of CL-Gab and CM-Gab. The granodiorite includes

quartz diorite and granodiorite. In this way, the co-genetic crystallization sequence between the gabbro and granodiorite groups is lithologically supported (Figure 3).

To verify such a fractional crystallization model, we examined the geochemical modeling using two folds: (1) mass balance calculation using major elements in the whole rocks and mineral compositions, and (2) the Rayleigh fractionation model adapting the results of mass balance calculation. The partition coefficients are listed in Table S4. In consideration of phase and compositional changes of fractional minerals, we classified the fractionation process into three stages: Stages 1, 2, and 3. The results are shown in Table 2 for the mass balance calculation and Figure 10 for the Rayleigh fractionation model. In conclusion, the formation of the granodiorite can be explained by the fractional crystallization process from basaltic magmas.

**Table 2.** Results of mass balance calculation using the whole-rock and mineral compositions for Fn-gabbro and the granodiorite.

| Stage 1 | Parent 17032305 | Daughter 16042904 | Fractionated Minerals | | | | calc. | diff. |
|---|---|---|---|---|---|---|---|---|
| | | | Pl | Opx | Cpx | Ol | | |
| fraction | 1.0000 | 0.3752 | 0.3044 | 0.1539 | 0.1245 | 0.0421 | | |
| $SiO_2$ | 49.26 | 53.24 | 45.18 | 53.01 | 52.88 | 35.92 | 49.26 | −0.06 |
| $TiO_2$ | 0.47 | 1.12 | 0.01 | 0.23 | 0.30 | 0.01 | 0.47 | −0.03 |
| $Al_2O_3$ | 17.32 | 18.05 | 34.88 | 1.16 | 1.32 | 0.01 | 17.32 | −0.06 |
| $Fe_2O_3$ | 8.54 | 8.95 | 0.18 | 19.16 | 7.37 | 33.91 | 8.54 | −0.07 |
| MgO | 8.08 | 3.76 | 0.03 | 23.81 | 15.51 | 28.89 | 8.08 | −0.05 |
| CaO | 11.12 | 7.65 | 18.31 | 1.19 | 21.81 | 0.03 | 11.12 | −0.05 |
| $Na_2O$ | 1.38 | 2.86 | 1.00 | 0.02 | 0.14 | 0.01 | 1.38 | −0.01 |
| $K_2O$ | 0.28 | 1.00 | 0.01 | 0.00 | 0.00 | 0.00 | 0.28 | −0.10 |
| | | | | | | | Sum of squares of residuals = | 0.03 |

| Stage 2 | Parent 16042904 | Daughter 16081401A | Fractionated Minerals | | | | calc. | diff. |
|---|---|---|---|---|---|---|---|---|
| | | | Pl(core) | Hb*(rim) | Hb(core) | Ilm | | |
| fraction | 1.0000 | 0.3446 | 0.4037 | 0.1506 | 0.0886 | 0.0126 | | |
| $SiO_2$ | 53.24 | 58.92 | 52.48 | 51.12 | 45.70 | 0.03 | 53.24 | 0.00 |
| $TiO_2$ | 1.12 | 0.72 | 0.01 | 0.08 | 1.45 | 51.73 | 1.12 | 0.06 |
| $Al_2O_3$ | 18.05 | 15.69 | 29.71 | 0.54 | 6.90 | 0.01 | 18.05 | 0.12 |
| $Fe_2O_3$ | 8.95 | 6.50 | 0.14 | 30.30 | 18.53 | 42.14 | 8.95 | −0.03 |
| MgO | 3.76 | 2.62 | 0.01 | 11.18 | 10.48 | 0.10 | 3.76 | 0.28 |
| CaO | 7.65 | 5.54 | 11.85 | 1.57 | 10.40 | 0.13 | 7.65 | −0.12 |
| $Na_2O$ | 2.68 | 2.75 | 4.59 | 0.08 | 1.21 | 0.02 | 2.68 | −0.04 |
| $K_2O$ | 1.00 | 2.17 | 0.15 | 0.02 | 0.66 | 0.01 | 1.00 | 0.12 |
| | | | | | | | Sum of squares of residuals = | 0.13 |

| Stage 3 | Parent 16081401A | Daughter 16042001 | Fractionated Minerals | | | | calc. | diff. |
|---|---|---|---|---|---|---|---|---|
| | | | Pl(rim) | Hb(rim) | Bt | Qz | | |
| fraction | 1.0000 | 0.3251 | 0.3225 | 0.1264 | 0.1262 | 0.0998 | | |
| $SiO_2$ | 58.92 | 65.08 | 54.89 | 48.09 | 34.58 | 100.00 | 58.92 | −0.04 |
| $TiO_2$ | 0.72 | 0.38 | 0.02 | 0.70 | 4.89 | 0.00 | 0.72 | −0.12 |
| $Al_2O_3$ | 15.69 | 13.82 | 28.15 | 5.52 | 14.04 | 0.00 | 15.69 | −0.05 |
| $Fe_2O_3$ | 6.50 | 3.34 | 0.15 | 18.79 | 22.72 | 0.00 | 6.50 | 0.14 |
| MgO | 2.62 | 1.18 | 0.01 | 10.79 | 9.23 | 0.00 | 2.62 | −0.31 |
| CaO | 5.54 | 3.09 | 10.24 | 10.60 | 0.69 | 0.00 | 5.54 | −0.08 |
| $Na_2O$ | 2.75 | 2.62 | 5.54 | 0.73 | 0.13 | 0.00 | 2.75 | 0.07 |
| $K_2O$ | 2.17 | 3.69 | 0.18 | 0.35 | 8.05 | 0.00 | 2.17 | −0.17 |
| | | | | | | | Sum of squares of residuals = | 0.17 |

Ol, olivine; Opx, orthopyroxene; Cpx, clinopyroxene; Pl, plagioclase; Hb*, tremolitic hornblende; Hb, hornblende; Bt, biotite; Ilm, ilmenite; Qz, quartz.

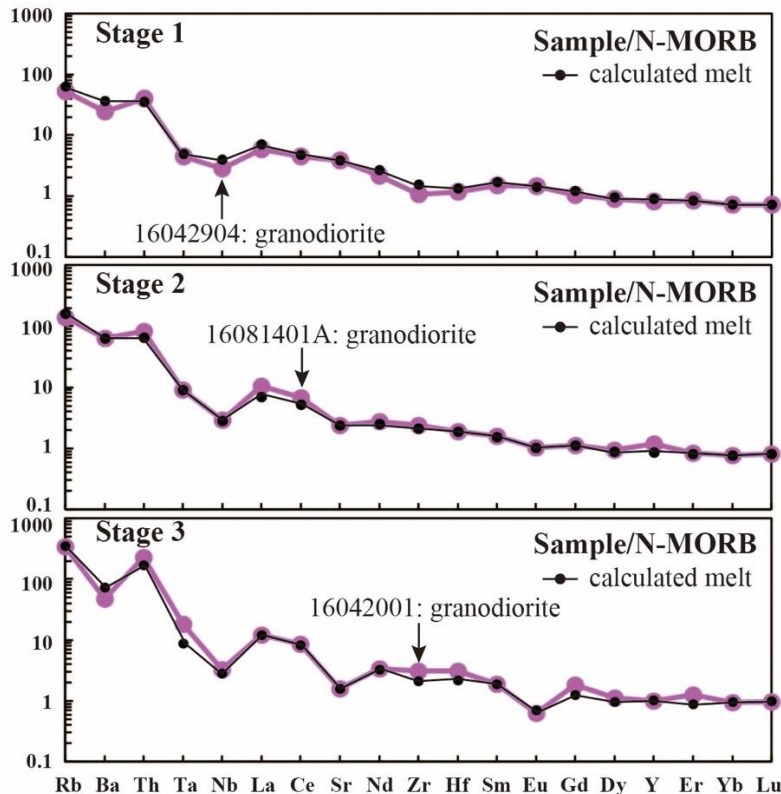

**Figure 10.** Results of fractional crystallization by the Rayleigh fractionation model using the results of mass balance calculation and partitioning coefficient. Normalized values of C1 chondrite are after the published paper [14]. The partitioning coefficients are listed in Table S4. The calculated magmas are identical with the granodiorite, suggesting that the granodiorite was differentiated from Fn-Gab as the parental magma.

### *5.2. Petrogenesis of Granite Magma*

Muscovite and garnet are commonly found in peraluminous granites that are generally formed by assimilation and/or partial melting of pelitic rocks for their magmatic genesis [20,21]. The granite from the study area includes the pelitic gneiss of the Ryoke metamorphic rock. The Sr–Nd isotopic compositions of the granite exhibit negative $\varepsilon$NdI values but are different from those of the pelitic gneisses (Figure 8). Therefore, the partial melting of the pelitic gneisses is ruled out to produce granite magma.

The open system (assimilation, mixing) processes were a candidate for the petrogenesis of a peraluminous granitic magma [21] because the differences in the SrI and NdI values in the granites could be attributed to the degree of assimilation rates or a mechanical mixture of the pelitic gneiss. According to Figure 8, the granite magma can be thought to be produced by the assimilation and fractional crystallization (AFC) process, of which the granodiorite and pelitic gneisses correspond to a parental magma and contaminant rocks, respectively. We adapted the AFC model calculation using the granodiorite with the highest $\varepsilon$NdI values as the parental magma and two pelitic gneisses as contaminants for the endmember compositions. The results of the AFC model calculation are shown in Figure 11. The blue and red lines indicate the estimated trends for the AFC process with the different contaminants. The two samples of granite are plotted out of these trends on the Sr–SrI diagram (Figure 11) although the granite can reproduce the Nd isotopic compositions by the AFC process (Figure 11). The off-trend Sr and SrI values of the granites (even if the Nd values can be explained by the AFC model) indicate that the AFC process cannot produce the granite magma. On the other hand, the mechanical mixing of the granodiorite and pelitic gneiss (contaminant 1) may produce the granite (Figure 11). The contaminant 1 pelitic gneiss and the parent granodiorite have $SiO_2$ contents with 63 wt%

and 59 wt%, respectively. If the SrI values of granite were attained by the mechanical mixture between the granodiorite and the contaminant 1 pelitic gneiss, the SiO$_2$ contents of the granite should be less than 63 wt%. The SiO$_2$ content of granites shows more than 67 wt% (Figure 6), thus the granite cannot be produced by the mechanical mixture between the granodiorite and the pelitic gneiss.

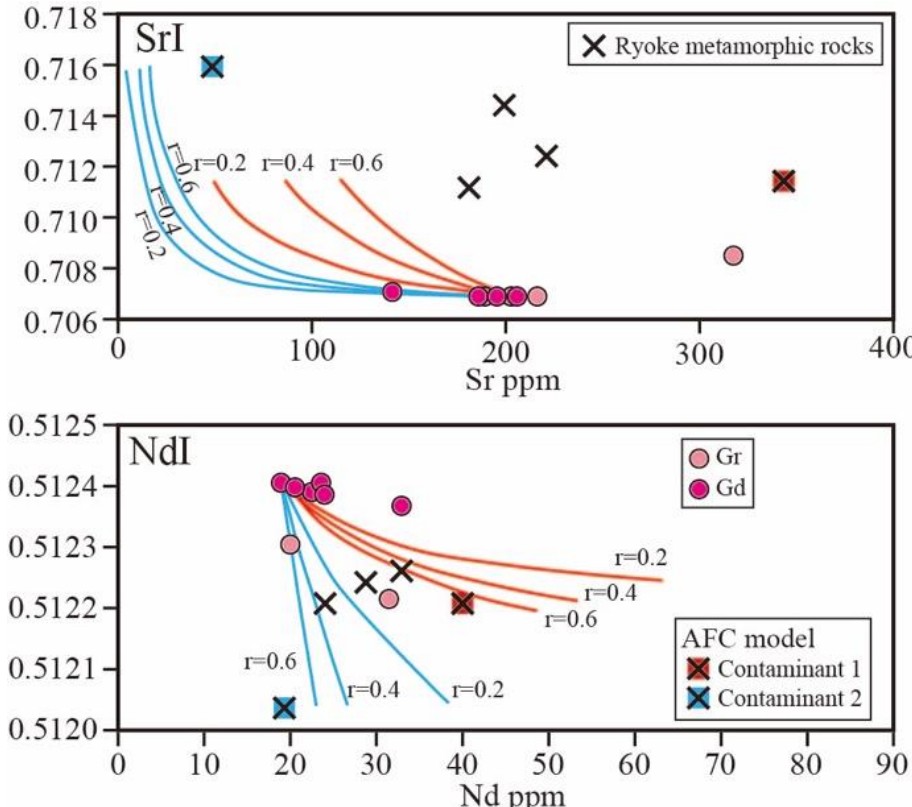

**Figure 11.** The results of assimilation and fractional crystallization (AFC) model. The granodiorite and the Ryoke pelitic gneiss correspond to a parental magma and contaminants, respectively. The "r" values indicate the ratio of assimilation/fractionation. The blue and red lines indicate the calculated AFC trends using the different contaminants. In Sr–SrI diagram, the granite deviates significantly from these lines. Therefore, the granite cannot be reproduced by the AFC process. See the text for detailed explanation.

The Sr–Nd isotopic compositions of the granite are plotted within the field of the Cretaceous, approximately 100 Ma, plutonic rocks from the Ryoke plutono–metamorphic complex (Figure 8). Previous works reported that the peraluminous granitic magmas can be produced by partial melting of igneous rocks with metaluminous compositions [15,22–26]. To test the generation of granite magma, we addressed the partial melting of the plutonic rocks from the Ryoke plutono–metamorphic complex because the Ryoke plutonic rocks were already emplaced in the middle to lower crust at c. 90 Ma in this area [15,27,28]. We chose two samples for the starting materials for the source of granitic magma because of the following reasons: (1) the Ryoke plutonic rocks have a compositional range from gabbro to granite, and (2) the chemical compositions of the two samples resemble those of the starting materials of the melting experiments that were conducted under the middle to lower crust conditions [20,21]. Table 3 shows the chemical compositions of the Ryoke plutonic rocks as starting materials.

**Table 3.** The chemical compositions of the Ryoke plutonic rocks as source rocks of the granite magma.

| (wt%) | SiO$_2$ | TiO$_2$ | Al$_2$O$_3$ | Fe$_2$O$_3$ | MnO | MgO | CaO | Na$_2$O | K$_2$O | P$_2$O$_5$ |
|---|---|---|---|---|---|---|---|---|---|---|
| A | 67.58 | 0.56 | 15.30 | 3.97 | 0.06 | 0.95 | 3.81 | 2.50 | 2.55 | 0.08 |
| B | 51.22 | 0.91 | 17.63 | 9.14 | 0.17 | 5.28 | 8.55 | 3.10 | 1.49 | 0.13 |

| (ppm) | Rb | Ba | Nb | Sr | Y |
|---|---|---|---|---|---|
| A | 115 | 562 | 13 | 238 | 11 |
| B | 68 | 262 | 5.3 | 240 | 18 |

| (ppm) | La | Ce | Nd | Sm | Eu | Gd | Dy | Er | Yb | Lu |
|---|---|---|---|---|---|---|---|---|---|---|
| A | 17 | 34 | 16 | 3.4 | 1.1 | 3 | 2.4 | 1.2 | 1.1 | 0.2 |
| B | 13 | 26 | 13 | 3.1 | 0.9 | 3.1 | 3.5 | 2.0 | 1.9 | 0.3 |

Samples A and B are granodiorite (09032207B) and diorite (12050503), respectively. The data are quoted from the published paper [15].

In order to examine the above-mentioned scenario, model calculations were performed by the formulation of a batch melting equation using trace element compositions and partitioning coefficients. The Ryoke plutonic rocks, granodiorite (09032207B), and diorite (12050503) [15] are regarded as source rocks. We used the experimental results as the modal abundance of residual phases [22,23]. The calculated results are shown in the spider diagram (Figure 12). The calculated compositions are plotted within the compositional range of the granite from the SNPC (gray field). The chemical composition of granite could, therefore, be reproduced by partial melting of the plutonic rocks with diorite and granodiorite compositions from the Ryoke plutono–metamorphic complex.

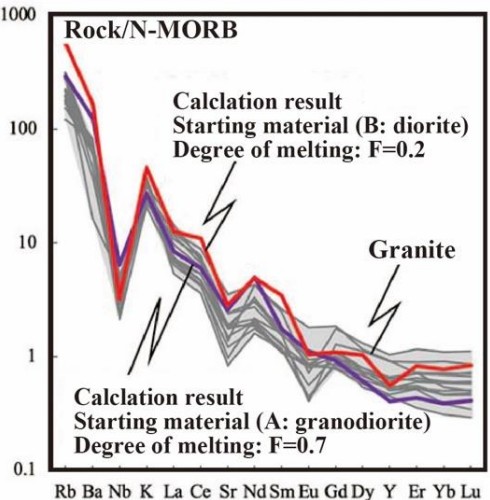

**Figure 12.** Spider diagram normalized to the primitive mantle [14] plotting the compositional ranges of the granite (gray field) and calculated granite magma (red and purple lines). The inferred granite magmas are plotted within the compositional field of granite. The granite magma can be produced by the partial melting of Ryoke plutonic rocks. See the text for further discussion.

### 5.3. Formation of Basaltic Magma and Cretaceous Mantle Dynamics

The Cretaceous magmatism in the west Chugoku district including Yamaguchi Prefecture occurred from c. 100 Ma to 87 Ma [29], and the peak magmatic stage appeared from 95 to 90 Ma [30]. The magmatic activity of the SNPC is identical with the peak stage. The nature of magmatic processes of the SNPC would, therefore, provides us an example for the growth and differentiation of continental crust along the active continental margin during the Cretaceous.

In the subduction zone, the wedge mantle changed to an enriched composition under the influence of slab-derived components to form a metasomatized mantle. Such mantle partially melts to produce hydrous basaltic magma. The An contents of plagioclase in

Co-gabbro are as high as An90 (Figure 5a). According to experimental results, high-An contents of plagioclase should be crystallized from a hydrous magma of more than 3 wt% in $H_2O$ [31–33]. Such hydrous magma deduced from the experimental results is consistent with the existence of hornblende within olivine and orthopyroxene as the early crystallized phase in the gabbro. Considering the existence of early crystallized hornblende and high-An plagioclase, the gabbro would be derived from hydrous magma. The hydrous basaltic magma generally shows low viscosity and also moves the melting temperature of the solidus to the lower temperature side [34]. This means that the hydrous magma can rise to the shallower level of crust. The gabbro from the SNPC, therefore, was emplacement at the shallower level in this region, then the complex had various compositions from the gabbro to granodiorite due to the process of fractional crystallization as previously mentioned.

Fn-Gab closely shows primitive compositions with Mg# as high as 69 (Table 3); however, the Ni contents are lower than 70 ppm [5]. Therefore, Fn-Gab magma is thought to be a slightly differentiated magma, which early crystallized Ol (containing Ni) was removed because the partitioning coefficient of Ni against olivine is high in basaltic magmas. We examined the calculation of primitive basaltic magma adopting the maximum olivine fractionation method [35] using Fn-Gab with Mg# = 69. The calculated results are listed in Table 4. Fn-Gab adding 7.5% olivine can coexist with mantle peridotite. The calculated primitive basaltic magma possesses Mg# = 75 with $SiO_2$ = 50.4 wt% (Table 4). As mentioned above, the basaltic magma studied here contains hornblende as the early crystallized phase. The $\varepsilon$NdI value of Fn-Gab shows negative (Figure 8). These geochemical data suggest that the basaltic magma should be derived from the metasomatized mantle wedge, where $\varepsilon$NdI isotopic composition is negative. The Nd isotopic compositions of Fn-Gab suggest that the mantle source was strongly modified by crustal materials with negative $\varepsilon$NdI values during the formation of basaltic magma.

**Table 4.** Inferred primitive compositions of basaltic magma deduced from the chemical compositions of Fn-gabbro using maximum olivine fractionation method [35].

|  | $SiO_2$ | $TiO_2$ | $Al_2O_3$ | FeO* | MnO | MgO | CaO | $Na_2O$ | $K_2O$ | $P_2O_5$ | Total | Mg# |
|---|---|---|---|---|---|---|---|---|---|---|---|---|
| Fn-Gab | 51.41 | 0.49 | 18.08 | 8.02 | 0.19 | 8.43 | 11.61 | 1.44 | 0.29 | 0.04 | 100.00 | 69 |
| Prim. Comp. | 50.42 | 0.45 | 16.44 | 8.35 | 0.17 | 12.01 | 10.55 | 1.31 | 0.27 | 0.04 | 100.00 | 75 |

Prim. Comp.: estimated primitive composition.

The magma activities in southwest Japan began at 105 Ma [29]. At that time, the episodic magmatism occurs in the Kinki district known as the Kyoto lamprophyre [29,36], which has similar petrologic characteristics to Fn-Gab [5]. The Kyoto lamprophyre includes hornblende as the early crystallized phase and can be divided into two groups, the positive and negative $\varepsilon$Nd groups [5,29]. Both lamprophyres were generated by the partial melting of mantle wedge under the garnet stability field because the Kyoto lamprophyre shows enrichment of LREE and depletion of HREE [29]. The heat source was thought to be an upwelling asthenosphere [29,36]. The REE patterns of Fn-Gab show almost flat in HREE, indicating the absence of garnet in the mantle source. The melting condition of Fn-Gab would, therefore, be shallower than the Kyoto lamprophyre. The magmatic activity of the SNPC occurred at c. 90Ma later than that of the Kyoto lamprophyre but the time of 90Ma was a peak magmatic activity in southwest Japan [30]. Therefore, the 90Ma magmatic activity in southwest Japan would occur due to the upwelling of the asthenosphere reaching up to the shallower level under the garnet unstable depth.

Finally, we consider the concept of magmatic plumbing systems, which is a view of arc magmatism in structure and composition. The hydrous basaltic magma is intruded into the crust and evolves in magma reservoirs where the magma differentiates by both closed system (fractionation) and open system (assimilation, mixing) processes, forming more felsic compositions [3]. In the SNPC, the basaltic magma evolved in an almost closed system. In any case, the product would be mafic cumulate like Co-gabbro and felsic melts, finally continuing their ascent to shallow levels to produce magma reservoirs [37,38]. The

cooling and solidification processes lead to the formation of plutonic bodies, while extracted magmas may complementarily erupt as andesites–rhyolites. In contrast, partial melting in the lower crust directly produces granitic melts. The source region occurs in the following processes, melting reaction, melt migration, and segregation. Finally, the extracted magmas rise towards emplacement sites, but uncommonly reach the surface [39,40]. The Cretaceous volcano–plutonic complexes ubiquitously exist in southwest Japan [30]. The Cretaceous volcanic rocks mostly show andesite–dacite–rhyolite in compositions but are little present in peraluminous dacites–rhyolites. This suggests that the SNPC studied here is one of the representative plutonic rocks related to the magmatic processes and crustal growth at the Cretaceous continental margin.

## 6. Conclusions

The Cretaceous SNPC in southwest Japan exemplifies magmatic processes in active continental margins. In this complex, the mantle-derived basaltic magma was differentiated during fractional crystallization processes to produce the granodiorite magma. In terms of the presence of early crystallized hornblende and high-An plagioclase (An $\geq$ 90), the basaltic magmas were thought to be hydrous. The underplated basaltic magma supplied thermal energy to the already existing intermediate to felsic crusts, which then underwent partial melting, giving rise to the granite magma. Mantle-derived magma contributes to the growth of the continental crust, whereas the partial melting of crustal rocks increases the geochemical maturity of the continental crust. The magmatic activity of the SNPC occurred at c. 90 Ma. This time is considered to be the peak of the Cretaceous magmatic activity in southwest Japan. The REE patterns in Fn-Gab suggest that the primitive basaltic magma was formed at shallow mantle depths at c. 90 Ma. At that time, the mantle wedge environment beneath southwest Japan is thought to have been hot due to the asthenosphere rising.

**Supplementary Materials:** The following supporting information can be downloaded at: https://www.mdpi.com/article/10.3390/min12060762/s1, Table S1: Representative mineral chemistry of the granite, Table S2: Whole-rock chemical analyses of the granite, Table S3: Sr–Nd isotopic analyses of the granite, Table S4: Partitioning coefficients using calculation of the Rayleigh fraction method for Fn-gabbro and granodiorite [41–78].

**Author Contributions:** S.K. and M.O.; fieldwork, chemical analyses, and writing—original draft preparation; M.N.; chemical analyses and writing—review and editing, A.K.; Sr–Nd isotopic analyses. All authors have read and agreed to the published version of the manuscript.

**Funding:** This research was funded by KAKENHI (Grant-in-Aid for Scientific Research) provided by the Japan Society for the Promotion of Science, grant number 15H03748 (M.O.), 18K03782 (M.N.), 18H01313 (A.K.).

**Acknowledgments:** We wish to thank Y. Morifuku for helping with EPMA works.

**Conflicts of Interest:** We declare that we have no known competing financial interests or personal relationships that could have appeared to influence the work reported in this paper.

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
