# Peer review of "Magmatic Processes of the Upper Cretaceous Susuma–Nagaho Plutonic Complex, Southwest Japan: Its Role on Crustal Growth and Recycling in Active Continental Margins"

_minerals, doi:10.3390/min12060762_

Round 1

Reviewer 1 Report

Dear authors

Thank you very much for your clearly presented paper. I embedded my comments to the pdf file

I only did not agree with on aspect and this is the pressure conditions calculated for hornblendes. One of the figures it seems like hornblendes are quite altered (Fig 4a) and the other one fig4b, it seems like an inclusion for me, but it can also be developed later in the magma chamber dynamics.

I resive you to re-calculate and check the pressure conditions of the hornblende, other than this the ms is quite decent and nicely written

best regards

Author Response

Dear reviewer 1,

Thank you for your fruitful comments. Your suggestion being “re-calculate and check the pressure conditions of the hornblende” is understandable but I calculated magmatic pressure using hornblende geobarometer by Molina et al. (2015) that equation can be accepted by many petrologists. Hornblende in figure 4a is an aggregation of fine-grained hornblende replacing an olivine grain. Therefore, we did not use the hornblende for pressure calculations. In contrast, hornblende in figure 4b is fresh and included in olivine; thus, such hornblende is crystallized at close to liquidus. We calculated the pressure conditions at early stage of crystallization of the basaltic magma. As we believe that the pressure estimation in this manuscript is logically reasonable, we don’t recalculate the pressure conditions at the time for hornblende crystallization during ascending of the basaltic magma. Please see the attached file.

with best regards,

Masaaki Owada

Reviewer 2 Report

This paper seeks to describe the relationships between three distinct igneous rock suites namely gabbro (representing mantle derived basaltic magma) granodiorite (arguably resulting from fractional crystallisation of basalt) and granite (the product of crustal melting promoted by the heat input from basaltic magma).The manuscript aims to establish the relationship between each of these suites.

The data on which the paper is based is in part already published but there is some new data particularly relating to the granitic rocks. Overall the manuscript is nicely presented but there needs to be a comprehensive edit of the English expression. The diagrams are nicely presented.

There is an acceptable discussion of the processes that link the different rock suites and of the fundamental contribution that different processes make toward growing continental crust.

Overall my recommendation is for acceptance subject to a rigorous edit of the English expression. In my opinion the manuscript represents a useful contribution to an understanding of the geology go southwest Japan and more widely a discussion of the way that continental crust developed. I have opted for major revision because the English expression really needs to be reviewed - perhaps more than at a minor level.

Author Response

Dear reviewer 2,

We would like to thank you for recognizing and appreciating the value of our research. Your comments are that the manuscript needs to be a comprehensive edit of the English expression. I already asked to the associate editor that your comments are not sufficient to revise the English expression because there were no specific pointers. We will follow the suggestions of the associate editor. Please see the attached file.

with best regards,

Masaaki Owada
